# Determination of Technological Properties and CRISPR Profiles of *Streptococcus thermophilus* Isolates Obtained from Local Yogurt Samples

**DOI:** 10.3390/microorganisms12122428

**Published:** 2024-11-25

**Authors:** Hatice Sevgi Coban, Dicle Olgun, İnci Temur, Muhammed Zeki Durak

**Affiliations:** 1Department of Food Engineering, Faculty of Chemical and Metallurgical Engineering, Yıldız Technical University, 34220 İstanbul, Türkiye; hsevgicoban@gmail.com; 2Department of Molecular Biology and Genetics, Faculty of Science and Letters, İstanbul Technical University, 34485 İstanbul, Türkiye; 3Department of Molecular Biology and Genetics, Faculty of Science and Letters, Yıldız Technical University, 34220 İstanbul, Türkiye

**Keywords:** *Streptococcus thermophilus*, yogurt, CRISPR, technologically important proteins

## Abstract

The aim of this study was to obtain data on Clustered Regularly Interspaced Short Palindromic Repeats (CRISPR) profiles of *Streptococcus thermophilus* (*S. thermophilus*) isolates resulting from acquired immune memory in addition to their technological starter properties for the selection of potential starter cultures from local yogurt samples. A total of 24 *S. thermophilus* isolates were collected from six local yogurt samples including Afyon/Dinar, Uşak, Konya/Karapınar, and Tokat provinces of Türkiye. Strain-specific CRISPR I-II-III and IV primers were used to determine the CRISPR profiles of the isolates. The isolates commonly had CRISPR II and IV profiles, while only one isolate had a CRISPR III profile. Polymerase chain reaction (PCR)-based and culture-based analyses were also carried out to obtain data on the technological properties of the isolates. The PCR analyses were performed for the *prtS* gene for protease activity, the *ureC* gene for urease enzyme, the *gdh* gene for glutamate dehydrogenase, the *cox* gene for competence frequency, the *csp* gene involved in heat-shock stress resistance of the isolates with specific primers. Culture-based analyses including antimicrobial activity and acid-production ability of the isolates were completed, and proteolytic and lipolytic properties were also screened. Native spacer sequences resulting from acquired immune memory were obtained for CRISPR IV profiles of yogurt samples from the Konya-Karapınar and Tokat provinces and CRISPR III profiles of yogurt samples from the Uşak province. In conclusion, our study results suggest that it is possible to select the isolates with the desired level of technological characteristics, prioritizing the ones with the most diverse CRISPR profiles and with native spacers for potential industrial application as starter cultures. We believe that this study provides data for further biological studies on the impact of centuries of human domestication on evolutionary adaptations and how these microorganisms manage survival and symbiosis.

## 1. Introduction

Studies evaluating potential starter cultures isolated from dairy products have provided extensive insights into their technological properties. These include acid-development potential, lactic acid quantification, exopolysaccharide locus sequence (EPS) isolation and purification, monosaccharide composition analysis of EPSs, antimicrobial activity, antibiotic susceptibility, and acid-forming capacity. However, there is a lack of comparable data based on Clustered Regularly Interspaced Short Palindromic Repeats (CRISPR) profiles that preserve the resistance capabilities of potential starters against phages, which are common in fermentation technology and are known to cause product losses both by lytic and lysogenic infection. In the context of optimizing starter cultures, modifications to adaptive immune mechanisms are extensively utilized. In this regard, it is of utmost importance to provide data obtained from microbial and genetic analysis methods. Identifying native spacer sequences will be particularly valuable for establishing a comprehensive collection of local potential starter cultures for the dairy industry.

Following the identification of the first phages specific to *Lactococcus lactis* strains by Whitehead and Cox in 1935, research in the dairy industry has increasingly focused on strategies to minimize losses in batch fermentations, comprising culture rotation, improved sanitation practices, and the use of starter cultures with enhanced phage resistance [1,2,3,4]. The isolation of bacteriophage-insensitive *Streptococcus thermophilus* (*S. thermophilus*) mutants (BIMs) is crucial for the dairy industry; however, mutations in phages that occur in response to CRISPR systems have been identified [5]. Non-CRISPR-mediated BIMs obtained by the transformation of industrial starter cultures with a constructed plasmid carrying antisense RNA, which binds to the targeted phage or phages’ mRNA and blocks translation, effectively inhibits phage replication and confers resistance to the starter culture. Studies conducted by McDonnell and colleagues reported that non-CRISPR-mediated BIMs obtained by the antisense RNA CRISPR-Cas silencing approach exhibited certain advantages in phage sensitivity tests compared to CRISPR-mediated BIMs derived from the same strain [6].

The CRISPR, known as adaptive immunity in bacteria and archaea, and the associated genes (*cas*), were first discovered in the *Escherichia coli* (*E. coli*) genome in 1987 during an analysis of genes involved in phosphate metabolism [7,8].

The sequences in the CRISPR region are usually similar to the sequences in bacteriophage (phage) genomes and plasmids that previously infected the microorganism, and the number of palindromic repeats determines the level of resistance [9,10,11,12,13].

The CRISPR profiles resulting from this acquired immune memory in microorganisms can also be used as a novel molecular characterization technique, as they vary at the species and subspecies levels [14,15,16]. The CRISPR systems are classified into six main types (types I–VI) and multiple subtypes. Type I, type II, and type V systems target deoxyribonucleic acid (DNA), and type VI systems target single-stranded ribonucleic acids (RNAs), while type III systems have the ability to cleave both DNA and RNA [11,17,18,19].

The Cas9 protein recognizes the species-specific protospacer adjacent motif (PAM) sequence adjacent to the protospacer in the DNA and prevents the expression of the bacteriophage DNA by cutting it. The difference in type I is that crRNA combines with cascade and Cas3 to cleave the phage DNA. In type III, crRNA works together with Cmr/Cas10 or Csm/Cas10, and it is not PAM-dependent. In addition, CRISPR IV is known to originate from plasmids [20]. Multiple CRISPR systems can coexist within a single microorganism.

To date, the CRISPR profiles and genes associated with technological traits in various lactic acid bacteria (LAB) have been thoroughly investigated [21,22,23,24]. The diversity of the adaptive immune mechanism in starter cultures, indicated by an extensive CRISPR profile, supports a range of essential technological properties. These include proteolytic activity, nitrogen and carbohydrate metabolism, transport systems, antimicrobial activity, resistance to various phages, flavor and aroma development, EPS production, urease activity, siderophore acquisition, and acid-forming capacity [21,22,25]. CRISPR gene modifications are widely applied in LAB for industrial use, and genetic modifications of adaptive immune mechanisms are also the subject of various microbiota studies for therapeutic purposes in LAB. Currently, CRISPR I, CRISPR II, CRISPR III, and CRISPR IV and their subtypes are commonly found in complete genome analyses and bioinformatic studies in LAB [16,26].

In the present study, to gain a better understanding of some technological properties of the isolates, polymerase chain reaction (PCR)-based analyses were conducted owing to their operational simplicity, and we completed the *prtS* gene, for protease activity, the *ureC* gene for urease enzyme, the *gdh* gene coding for glutamate dehydrogenase, the *cox* gene for frequency of competency, and the *csp* gene involved in strain heat-shock stress resistance and the *EPS* gene cluster, targeting the *EPS* by specific primers.

### 1.1. Protease (prtS Gene)

Casein degradation is one of the important stages affecting the product’s texture and taste. Similarly, targeting the proteolytic system can prevent the formation of an undesired bitter flavor caused by some particular peptides [27]. Proteases are necessary enzymes for the protein hydrolysis activity of the microorganisms. *S. thermophilus*, like other LAB, has its own proteolytic system. Proteases are found and have various roles in different steps of the proteolytic system.

### 1.2. Urease (ureC Gene)

Urease is an enzyme that catalyzes the hydrolysis of urea into ammonia and carbonic acid. First, urea is broken down into ammonia and carbamate; then, carbamate further decomposes into ammonia and carbonic acid. Due to the production of ammonia, the pH of the environment rises. Urease also plays a crucial role in nitrogen metabolism by assisting the usage of urea as a nitrogen source [28,29,30]. Quick acidification of milk during dairy fermentation can be significant to the quality of the final product. Lactic acid produced by LAB lowers pH and coagulates milk proteins to give the desired texture. However, as the ammonia produced by urea hydrolysis raises pH and slows down acidification, urease activity may prevent this process, leading to irregularities in the fermentation process and an undesirable effect on the finished product’s flavor and texture. Since they do not hinder milk acidification, urease-negative strains of *S. thermophilus* are therefore preferred in the dairy industry [28,29].

### 1.3. Glutamate Dehydrogenase (gdh Gene)

Glutamate dehydrogenase (gdh) is an enzyme that plays an important role in amino acid metabolism. This enzyme catalyzes the reversible reaction of converting glutamate into alpha-ketoglutarate. Therefore, GDH is involved in both the catabolic and anabolic processes. The transamination reaction is seen as crucial, particularly during cheese making [31].

### 1.4. Cold Shock Proteins (csp)

Many types of bacteria can grow at temperatures well below their optimum growth temperature. In some bacteria, cold shock proteins (*csp*s), which help bacteria adapt to cold conditions, are induced as the temperature drops. The *csp*s are proteins that make up a small, highly conserved chaperone family that bind to nucleic acids, ensuring transcription and translation functions correctly at low temperatures. Therefore, freeze survival capacity is an essential metric in the dairy industry [32]. The *S. thermophilus* is one of the most important LAB used in dairy processing plants.

### 1.5. Competence Frequency (cox Gene)

The presence of the *comX* alternative sigma factor activity is considered to be related to the conjugation ability of *Streptococci* [33]. Sigma factors play a key role in the initiation of transcription of second-class genes, also known as late genes, which are necessary for the DNA uptake and integration machinery.

To further diversify the data obtained in our study, we selected four additional technological analyses that would provide faster and beneficial results. Accordingly, antimicrobial activity and acid-production capacities of the isolates were examined, and their proteolytic and lipolytic properties were screened.

## 2. Materials and Methods

### 2.1. Microbial Analyses

#### 2.1.1. Isolation of *S. Thermophilus* Isolates

Six different local yogurt samples were collected from different provinces of Türkiye including Afyon/Dinar, Uşak, Konya/Karapınar, and Tokat provinces for this study. A total of 10 g of yogurt samples was mixed with 90 mL of sterile 0.1% (*w*/*v*) peptone water (Merck, Darmstadt, Germany). Dilutions from 10^−1^ to 10^−8^ were prepared, M17 Agar (Merck, Darmstadt, Germany) was cooled to 65 °C, and 1 mL of diluted samples from 10^−6^ to 10^−8^ was used for the double-layer pour-plate technique. After incubations at 42 °C for 48 h under anaerobic conditions, single colonies were picked from plates and subcultured three times until pure colonies were obtained. Purity was confirmed by colony morphology and microscopic observations. Parallel 20% (*v*/*v*) glycerol stocks were prepared for long-term preservation of cultures. The cultures were stored at −80 °C.

#### 2.1.2. Acid-Production Ability

The acid-production ability of the isolates was measured as previously described by Bulut et al., 2005, with minor modifications. Each isolate was incubated overnight, and 100 μL of culture was inoculated into 10 mL skim milk (Merck, Darmstadt, Germany) (10 g/L) and incubated at 42 °C [34]. Titratable acidity and pH measurements of each isolate were recorded for 2 mL of culture samples following incubation at 3, 6, and 9 h, respectively.

#### 2.1.3. Proteolytic and Lipolytic Activity

The proteolytic activity of the isolates was screened by using 1% skim milk powder, and 0.8 g/L nutrient agar (Merck, Darmstadt, Germany) was added [35]. Isolates with clear transparent haloes were recorded as positive proteolytic activity. Lipolytic activity of the isolates was detected by using 10 g/L tween 20 or tween 80 plates including 0.1 g/L CaCl_2_ and 0.8 g/L nutrient agar (Merck, Darmstadt, Germany). Isolates were spotted onto agar plates after incubation at 42 °C, and screening results at 72 h were recorded [36].

#### 2.1.4. Antimicrobial Activity

Antimicrobial activity of the isolates against *Bacillus cereus* CCM 99, *E. coli* ATTC 25922 was observed by a well-diffusion assay with minor modifications [37]. One hundred μL *E. coli* (10^7^ cfu/mL) were poured on Mueller–Hinton agar (Merck, Darmstadt, Germany). Wells were cut on the plates, and 100 μL of overnight cultures of *S. thermophilus* isolates was filled into the well and incubated at 42 °C for 48 h. Diameters of zones were recorded according to antimicrobial properties.

### 2.2. Genetic Analysis

#### 2.2.1. CRISPR Profiles of the Isolates

Bacterial DNA isolation of 24 isolates was completed by an Intron G-Spin DNA extraction kit (iNtRON Biotechnology, headquartered in Seongnam-si, Gyeonggi-do, Republic of Korea) with catalog number 17045. CRISPR I-II-III and IV profiles of the isolates were detected by using S. thermophilus strain-specific primers (Sentebiolab Biotechnology, Istanbul, Türkiye) [4,16] targeting repeat/spacer loci (Table 1). The PCR conditions of pre-denaturation were 5 min at 95 °C; 40 cycles were carried out for 1 min at 95 °C; the annealing procedure was carried out for 1 min at a suitable temperature for each CRISPR primer pair; the extension procedure was carried out for 1 min at 72 °C; and the final extension procedure was carried out for 10 min at 72 °C. The PCR products were separated by electrophoresis on agarose 3.0% gels (Thermo Fisher Scientific, Waltham, MA, USA). Following purification, they were sequenced (Medsantek, Istanbul, Türkiye) using the ABI 3500 XL Genetic Analyzer (Applied Biosystems, Carlsbad, CA, USA) with the BIG-DYE cycle sequencing kit (Thermo Fisher Scientific, Waltham, MA, USA). Acquired adaptive immunity is established by the positioning of spacer regions derived from exogenous nucleic acid (phage or plasmid) sources within the CRISPR region. Therefore, to quantify the number of native spacers of each isolate, sequences were aligned in BioEdit sequence alignment editor v7.2.5 (12 November 2013). To analyze the aligned sequence results, a web-based tool called CRISPR Finder (https://crisprcas.i2bc.paris-saclay.fr/CrisprCasFinder/Index, accessed on 1 May 2024) was used. The sequences were saved as native sequences, except when they matched the sequences previously recorded in the database.

#### 2.2.2. Technologically Important Proteins of the Isolates

The presence of genes encoding for technologically important proteins was analyzed using specific primers. For the presence of the *prtS* gene, coding for proteases involved in milk casein breakdown [38,39], the *ureC* gene related to urease enzyme [30,39]; the *gdh* gene, coding for GDH [39,40]; the *csp* gene, involved in the strain heat-shock stress resistance [32,39]; and the *comx* gene for frequency for competency [33] were selected (Table 2).

## 3. Results

### 3.1. Microbial Analyses

#### 3.1.1. Isolation of *S. thermophilus* Isolates

Twenty-four isolates from Afyon/Dinar, Uşak, Konya/Karapınar, and Tokat provinces of Türkiye using M17 agar plates were obtained. Colonies in spherical and oval morphology observed in the Petri dishes and cocci isolates with short- and long-chain structures were confirmed by microscopic observations. Parallel glycerol stocks were prepared for 24 purified isolates stored at −80 °C. A list of 24 *S. thermophiles* selected from 137 isolates obtained from six different yogurt samples from Afyon/Dinar, Uşak, Konya/Karapınar, and Tokat is presented in Table 3.

#### 3.1.2. Acid-Production Ability

*S. thermophilus* isolates in this study reduce the pH of the series to 3, 6, and 24 h, and they are limited acid producers (Table 4). Acid-production values of the isolates are presented in Table 4, and most of the isolates lowered the pH below 6 following 24 h incubation in this study. During all measurement periods, strain K31 increased the acidity the most, reaching 3.98. The slowest pH declines were as follows: K31 (6.52) and T6–9 (6.47) at 3 h; T6.9 (6.3) and T3.10 (6.22) at 6 h; and the B5 strain with pH values of 4.7 at 24 h.

#### 3.1.3. Proteolytic and Lipolytic Activity

Proteolytic activities were detected in all of our isolates, whereas lipolytic activity could not be observed for any of the isolates.

#### 3.1.4. Antimicrobial Activity

In our study, antimicrobial activity could not be observed against *Bacillus cereus* CCM 99 and *E. coli* ATTC 25922 for any of the isolates.

### 3.2. Genetic Analysis

#### 3.2.1. CRISPR Profiles of the Isolates

The isolates commonly had CRISPR II (20%) and IV (75%) profiles, while only one isolate had a CRISPR III (4%), and one of the isolates (4%) had both CRISPR II and CRISPR IV profiles. The CRISPR profiles of the isolates detected based on PCR results are listed in Table 5. According to sequencing results, most of the native spacers were obtained from the isolates with the CRISPR IV profile from the Konya province, except for one of the isolates with the CRISPR III profile isolated from the same province (Table 6).

#### 3.2.2. Technologically Important Proteins of the Isolates

All of the 24 isolates were subjected to PCR analysis for technologically important proteins, and the results are summarized in Table 7. The *Prts* (milk casein breakdown) gene PCR was positive for 20 isolates (83%). *Comx* (frequency of competency)-gene PCR was positive for 14 isolates (58%). For the *gdh* gene, PCR was positive only for 2 (8%) isolates. Any of the isolates were PCR positive for the *csp* (heat-shock stress resistance) gene and *urease* gene (Table 7).

## 4. Discussion

In recent years, fermented products have become increasingly important in nutrition. Originally developed to preserve foods for extended periods, fermentation methods have evolved and diversified across cultures to enhance the nutritional value of foods, improve sensory qualities, and incorporate culturally specific techniques for culinary purposes [41]. Currently, yogurt is used as an easily accessible protein source and particularly to protect the intestinal microbiota [39,42]. Therefore, conducting microbial and genetic research on LAB is crucial for developing starter culture libraries. It is necessary to acquire more data for the technological properties and CRISPR profiles of LAB occurring by natural horizontal gene transfer that defines how the resistance abilities against phages, which are common in fermentation technology and known to cause product losses, are preserved [9,43].

In terms of technological characteristics, acidifying capacity, proteolytic and lipolytic activities, and antimicrobial activity were investigated in a total of 24 *S. thermophilus* isolates collected from six local yogurt samples from the Afyon/Dinar, Uşak, Konya/Karapınar, and Tokat provinces of Türkiye (Table 3). The acidifying capacity of LAB is important for the production of aroma and texture formation in dairy products. Industrial yogurt production is based on the symbiosis between *S. thermophilus* and *Lb. bulgaricus*, an equilibrium ratio adjusted to 1:1 or 2:3 to distribute the colloidal calcium phosphate to complete dissolution, dissolved caseins, and a yogurt gel matrix obtained within 282 to 330 min [44,45]. Acidifying capacities of the isolates selected for the study were investigated, and it was confirmed that all isolates lowered the pH below 6, and strain K31 was recorded as increasing the acidity the most, reaching 3.98 (Table 4). Both intracellular and extracellular enzymes including proteolytic and lipolytic activity have an important role in texture and aroma development during dairy fermentation processes. Various studies have shown that *S. thermophilus* strains typically exhibit limited protease and lipase production, resulting in minimal contribution to lipid and milk-protein hydrolysis. This becomes more evident when compared to other LAB, such as *Lactobacillus* or *Lactococcus* species, which demonstrate higher protease and lipase activity. All isolates were screened for extracellular proteolytic and lipolytic activity. All of the isolates had extracellular proteolytic activity, but extracellular lipolytic activity could not be detected as was consistent with the literature. The antimicrobial activities of the isolates were investigated to evaluate their inhibitory effects against pathogenic microorganisms or other bacteria that cause food spoilage. It is well documented in the literature that the antimicrobial properties of *S. thermophilus* are rarely observed. In line with previous studies, the isolates in this study were unable to inhibit *Bacillus cereus* CCM 99 and *E. coli* ATCC 25922.

PCR-based methods were also used for the screening of technological properties including protease activity, urease enzyme, glutamate dehydrogenase, competence frequency, heat-shock stress resistance, and *exopolysaccharide* genes with specific primers. Most of the isolates in this study were recorded as *prtS* positive (79%). The enzyme in *S. thermophilus* is encoded by the gene cluster *ureABC*, which also includes accessory genes *ureEFGD* and *ureI* that control the biogenesis of urease. Although urease is expressed by the majority of *S. thermophilus* strains, some urease-negative mutants have been created that show faster acidification in milk, which makes them more appropriate for industrial yogurt production [29,30,46]. The UREC-negative status of all our isolates further supports their ability to regulate pH during fermentation. In the production of dairy products, the GDH enzyme contributes to protein breakdown and the production of aromatic compounds. LAB, particularly during the fermentation process, help the production of free amino acids from milk proteins and the formation of compounds that play an important role in product taste and flavor. Isolates *K31* and *H35* were found to be promising LAB candidates for further dairy applications due to both technological and probiotic properties.

According to Karaman, mutations in the *comX* gene have a strong relationship with competence frequency and conjugation capacity [33]. The *comX* gene is critical in terms of increasing quality and efficiency in yogurt- and cheese-production processes where S. thermophilus and Streptococcus salivarius bacteria are used, as it provides natural competence [47]. The potential starter properties of the isolates are confirmed by the presence of *comX* genes, which are responsible for DNA preservation and successful genetic transformation in 60% of the isolates demonstrating natural conjugation capacity.

However, the presence of cold shock protein *(csp)* genes seems to have adapted the bacteria to cope with stress during commercial production [48]. Several studies have also tested the freezing tolerance of *S. thermophilus* after a cold-shock treatment. Overall, the results indicate that different strains and treatment conditions (temperature and time) yield variable results [49]. Nonetheless, further genomic and proteomic studies are needed to better understand their potential for cold adaptability. Cold stress is thought to be critical for the survival of frozen starter cultures and for fermentation occurring at low temperatures [50]. None of the isolates were detected as *csp*-positive in this study. The relationship between ecological factors and the evolution of microbial immune strategies and the increasing diversity of known prokaryotic defense systems has been investigated [51,52]. Based on genetic analysis, the CRISPR profiles of the isolates were investigated, and five of the isolates had CRISPR II (20%), 18 of the isolates had a CRISPR IV (75%) profile, and one of the isolates had a CRISPR III (4%) profile. The coexistence of multiple CRISPR systems within a single microorganism enhances the starter culture potential of these isolates. In particular, one of the isolates (4%) named as HC1.3 was found to be a promising candidate LAB for further dairy applications due to harboring both CRISPR II and IV genes. Based on the sequence results, native spacers were recorded for the isolates with CRISPR IV and CRISPR III profiles. Native spacer sequences were not recorded for the isolates with the CRISPR II profile. The alignment of spacer sequences in isolates with the CRISPR II profile to existing database entries indicates pre-existing acquired immunity against specific phages [1]. However, spacer sequences that do not match any database entries do not provide conclusive evidence that these isolates have encountered unknown or mutant phages. Instead, this more likely suggests that the corresponding phage sequences have yet to be entered into the database, as it is evident that more data on CRISPR sequences in LAB is urgently needed. The plasmid-encoded nature of CRISPR IV suggests that this could be the underlying reason for its widespread occurrence in the isolates. In light of these data, it is possible to select the isolates with the desired level of technological characteristics, prioritizing the ones with the most diverse CRISPR profiles and with native spacers for potential industrial application as starter cultures. We did not observe a specific CRISPR profile in strains with a higher number of positive technological characteristics, indicating that the selected traits are not directly correlated.

We believe that this study provides data for further biological studies on the impact of centuries of human domestication on evolutionary adaptations and how these microorganisms manage survival and symbiosis.

In this study, microbial and genetic analyses were conducted to evaluate the starter potential of isolates from different provinces of Türkiye. While culture-based screening methods are useful for the initial differentiation of isolates, whole-genome analysis demonstrates potential as a precise and feasible solution for obtaining more detailed data in future studies.

## Figures and Tables

**Table 1 microorganisms-12-02428-t001:** List of primers selected for CRISPR profile PCR analysis [4,16].

Primer	Sequence (5′-3′)	Target Gene
CR1-fwd (yc70)	TGCTGAGACAACCTAGTCTCTC	CRISPR I
CR1-rev	TAAACAGAGCCTCCCTATCC	
CR2-fwd	TTAGCCCCTACCATAGTGCTG	CRISPR II
CR2-rev	TTAGTCTAACACTTTCTGGAAGC	
CR3-fwd	CTGAGATTAATAGTGCGATTACG	CRISPR III
CR3-rev	GCTGGATATTCGTATAACATGTC	
CRP4-fwd	GATTCAGTTCCTCATAGAGC	CRISPR IV
CRP4-rev	GACCTCAACCAATCGATTG	

**Table 2 microorganisms-12-02428-t002:** List of primers selected for screening technologically important proteins of the isolates [33,39].

Primer Name	Amplicon Size (bp)	Primer DNA Sequence	Reference
ComXF	497	ATGGAACAAGAAGTTTTTGTTAAGGC	[33]
ComXR		TCAGTCTTCTTCATTACATGGATCAA	
ureC-F	1340	GGGGATAGCGTACGTCTTGG	[39]
ureC-R		TCAGCCAGCATCACCCATAACAC	
cspF	119	TTATTACCTCTGAAGATGG	[39]
cspR		ACGTTGACCTACTTCAACAT	
gdhf	1353	TTGCCAAAGCTTCATGACTG	[39]
gdhr		ACATGGGAAAGCCAAGTCAG	
prtsF	1501	GTGAGGCTTTGGCAGCTAAC	[39]
prtsR		TCGCGATATAGACCGGATTC	

**Table 3 microorganisms-12-02428-t003:** List of the isolates.

Isolate Number	Sampling Location
B5	Uşak
B6	Uşak
B8	Uşak
T2.1	Konya-Karapınar
T3.10	Konya-Karapınar
T3.1	Konya-Karapınar
T3.9	Konya-Karapınar
T6.9	Konya-Karapınar
T4.1	Konya-Karapınar
T6.5	Konya-Karapınar
T6.7	Konya-Karapınar
T6.4	Konya-Karapınar
T6.13	Konya-Karapınar
T6.14	Konya-Karapınar
T6.1	Konya-Karapınar
T3.2	Konya-Karapınar
Tokat 2.7	Tokat
Tokat 3.7	Tokat
K31	AFYON Dinar
KF30	AFYON Dinar
HC1.3	AFYON Dinar
A142	AFYON Dinar
H35	AFYON Dinar
AL	AFYON Dinar

**Table 4 microorganisms-12-02428-t004:** pH-reduction abilities of the isolates.

Isolate Number	3 h	6 h	24 h
B5	6.09	5.72	4.7
B6	6.29	5.79	4.5
B8	6.28	5.68	4.41
T2.1	6.4	6.18	4.11
T3.10	6.4	6.22	4.58
T3.1	6.16	5.46	4.07
T3.9	6.28	5.66	4.18
T6.9	6.47	6.3	5.1
T4.1	6.42	6.1	4.11
T6.5	6.31	5.83	4.04
T6.7	6.34	5.93	4.08
T6.4	6.23	5.65	4.37
T6.13	6.32	6.13	4.72
T6.14	6.13	5.52	4.35
T6.1	6.22	5.56	4.04
T3.2	6.29	5.85	4.09
Tokat 2.7	6.08	5.58	4.5
Tokat 3.7	6.31	5.86	4.06
K31	6.52	6.16	3.98
KF30	6.03	5.67	4.39
HC1.3	6.13	5.67	4.47
A142	6.1	5.58	4.39
H35	5.89	5.44	4.11
AL	6.09	5.6	4.48

**Table 5 microorganisms-12-02428-t005:** CRISPR profiles of the isolates.

Isolate Number	Location	CRISPR 1	CRISPR 2	CRISPR 3	CRISPR 4
B5	Uşak	−	−	−	+
B6	Uşak	−	−	+	−
B8	Uşak	−	+	−	−
T2.1	Konya-Karapınar	−	−	−	+
T3.10	Konya-Karapınar	−	−	−	+
T3.1	Konya-Karapınar	−	−	−	+
T3.9	Konya-Karapınar	−	−	−	+
T6.9	Konya-Karapınar	−	−	−	+
T4.1	Konya-Karapınar	−	−	−	+
T6.5	Konya-Karapınar	−	−	−	+
T6.7	Konya-Karapınar	−	−	−	+
T6.4	Konya-Karapınar	−	−	−	+
T6.13	Konya-Karapınar	−	−	−	+
T6.14	Konya-Karapınar	−	−	−	+
T6.1	Konya-Karapınar	−	−	−	+
T3.2	Konya-Karapınar	−	−	−	+
T2.7	Tokat	−	−	−	+
T3.7	Tokat	−	−	−	+
K31	Afyon-Dinar	−	+	−	−
KF30	Afyon-Dinar	−	+	−	−
HC1.3	Afyon-Dinar	−	+	−	+
A14.2	Afyon-Dinar	−	−	−	+
H35	Afyon-Dinar	−	+	−	−
AL	Afyon-Dinar	−	−	−	−

**Table 6 microorganisms-12-02428-t006:** Isolates with native spacer sequence.

Isolate Location, Number	CRISPR Profile	Number of Native Spacers
Usak, Isolate B6	CRISPR 3	13 native spacer sequences
Konya-Karapınar, Isolate T6.7	CRISPR 4	11 native spacer sequences
Konya-Karapınar, Isolate T6.14	CRISPR 4	13 native spacer sequences
Konya-Karapınar, Isolate T6.9	CRISPR 4	6 native spacer sequences
Tokat, Isolate T3.7	CRISPR 4	2 native spacer sequences
Konya-Karapınar, Isolate T6.13	CRISPR 4	9 native spacer sequences

**Table 7 microorganisms-12-02428-t007:** PCR results obtained from technologically important proteins.

Isolate Number	*comx*	*ureC*	*prts*	*gdh*	*csp*
B5	−	−	+	−	−
B6	−	−	+	−	−
B8	−	−	+	−	−
T2.1	−	−	+	−	−
T3.1	−	−	+	−	−
T3.2	−	−	+	−	−
T3.9	−	−	+	−	−
T3.10	−	−	+	−	−
T4.1	−	−	+	−	−
T6.1	−	−	−	−	−
T6.4	+	−	−	−	−
T6.5	+	−	+	−	−
T6.7	+	−		−	−
T6.9	+	−	−	−	−
T6.13	+	−	+	−	−
T3.14	+	−	+	−	−
Tokat 2.7	+	−	−	−	−
Tokat 3.7	+	−	−	−	−
K31	+	+	+	+	−
KF30	+	−	+	−	−
HC1.3	+	−	+	−	−
A14.2	+	−	+	−	−
H35	+	−	−	+	−
AL	+	−	+	−	−

## Data Availability

The original contributions presented in this study are included in the article. Further inquiries can be directed to the corresponding author.

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
