# Peer review of "Determination of Technological Properties and CRISPR Profiles of Streptococcus thermophilus Isolates Obtained from Local Yogurt Samples"

_microorganisms, 2024, doi:10.3390/microorganisms12122428_

Round 1

Reviewer 1 Report

Comments and Suggestions for Authors

The article addresses a highly interesting topic that can be applied to both the selection of starters in the food sector and other fields of microbiology. Thus, the study holds significant importance for the area. However, some critical aspects require improvement.

The methodology for species-level identification of the S. thermophilus isolates is not described. It is also recommended that the article clearly highlight what was newly achieved, as it is not evident how these results are contributing to the field of study. 

The abstract outlines the main results, but it needs refinement to better explain the connection between the CRISPR profile and the various analyses of the technological properties of the starters. Is there any evidence that the incidence of CRISPR sequences directly proportional to the complementary analyses of gene presence and antimicrobial activity? This point is unclear and would be great to know. 

In the introduction, I suggest adding a sentence to mention that in-depth study of CRISPR sequences has implications for gene editing, which is also highly relevant in food technology. Lines 98 to 105 briefly touch upon this and could be moved earlier in the introduction. This suggestion stems from the fact that the first association with CRISPR is often gene editing, even though that is not the focus of this work.

Moreover, it would be goof to include references to other studies that have characterized CRISPR sequences to access immune response in bacterial strains.

Line 18: Cite "Turkey" in English instead of the original language, as well as in other instances where it appears. This will contribute to access your nation results in academia research. 

In the introduction section, from lines 27 to 49, there are no theoretical references. Please include citations.

Lines 67-68: lactic acid bacteria are part of the bacterial group. This sentence needs to be revised, as it is unclear and requires proper referencing.

Lines 74, 80, and 81: It is suggested to select the most promising and current studies for references, as there are many included, and some may be more relevant than others.

Line 107: Use only the abbreviation "LAB," as it was previously cited in the introduction.

Overall, the introduction could be better structured to highlight the relevance of all the tests conducted for determining the technological properties of the strain, citing other works in the field that have been published for the same type of analysis regarding strain immunity.

The sections of the introduction discussing genes are interesting but somewhat exhaustive. The research conducted is highly significant, and a lengthy section on genes is not the primary focus. Therefore, I suggest summarizing it.

Lines 212, 303, 343, 359, 387: Correct double spacing.

Lines 217 and 273: Species names should be italicized.

Lines 221 and 222: Use superscript for "-1" and "-8" 

The materials and methods section does not include how the strains were identified. Morphological classification alone is insufficient for species-level identification. It is necessary to describe the identification protocol for the isolated strains. Furthermore, it is expected that the items in the materials and methods follow a similar order to that described in the introduction and/or results and discussion sections. Please, check. 

Line 228: The reference does not adhere to the journal's format.

Lines 238 and 244: Correct the formatting of the "2" in CaCl₂ and the "7" in 10⁷ cfu/mL. Please, review this throughout the text.

Line 242:  Bacillus cereus CCM 99 AND E. coli ATCC 25922.

Were the proteolytic, lipolytic, antimicrobial activities, and CRISPR profile analyses conducted following any previously described protocol? Please cite accordingly.

Lines 249, 264, 340: Use "isolates."

Line 252: I suggest placing all tables immediately after they are cited to facilitate reading.

Lines 274-280: Are selective media and morphological characteristics sufficient for species-level identification? What evidence supports this conclusion?

Lines 283-286: The sentence is confusing; please rephrase. Also, the reference does not follow the journal's standards.

Line 296: S. thermophilus.

Line 300: Were tests for antimicrobial activity conducted against S. aureus and P. aeruginosa? These are the most commom for this essay, besides E. coli

Lines 331-335: The sentence is unclear; please reconstruct it.

Lines 343-344: WAS positive.

Line 350: References in Table 2 are not in the correct journal format.

Line 352: Standardize tables font size.

Lines 357-362: Add a reference.

Line 364: Use the abbreviation "LAB."

Lines 378-380: The sentence raises the question of why these analyses have not yet been conducted if are important. I suggest omitting this since it was not performed. Instead, you may state that further exploration of EPS production capabilities could be achieved through primer analysis X. 

Line 381: Check whether "Streptococcal" should be italicized and standardize it.

Regarding the discussion and conclusion section, it appears to be lacking in depth. The main conclusion is unclear. The data need to be compared with other studies on this topic. It is not evident how your study has genuinely advanced the selection of starter cultures. You have significant results, but they need better discussion. For example, were any specific CRISPR profile present in strains with a higher number of positive technological characteristics? Are there other LAB strains with the same profiles? If so, which ones? Was this the expected outcome for your group of study? What was the most significant conclusion of the article following this characterization of the isolates? Which analyses would be most promising for selecting the most suitable isolates for application as starters?

  Comments on the Quality of English Language

Overall, the English language in the article could be improved by increasing formality, enhancing the use of connectors, and improving sentence transitions to elevate the quality of the material. Also, avoid the use of colloquial language. 

Lines 362-363: In this sentence, "enriches our table" and "intestinal flora" should be written in more formal English. The same applies to lines 381-384.

Author Response

November 18th, 2024

Dear Editor,

On behalf of my colleagues, I would like to thank you and the reviewers for their efforts and comments which led us to improve the scientific quality of our manuscript entitled “Determination of technological properties and CRISPR profiles of Streptococcus thermophilus isolates obtained from local yogurt samples” and for the opportunity to address the reviewers’ comments.

We thank the reviewers for their constructive comments. All relevant amendments were made accordingly as highlighted in the main manuscript in yellow color. A point-by-point response to the reviewers’ comments has been presented below.

Kindly submitted for your consideration for publication.

Yours sincerely,

Prof. Dr. Muhammet Zeki Durak

Department of Food Engineering,

Faculty of Chemical and Metallurgical Engineering,

Yıldız Technical University, İstanbul, TÜRKİYE

Comments 1: The article addresses a highly interesting topic that can be applied to both the selection of starters in the food sector and other fields of microbiology. Thus, the study holds significant importance for the area. However, some critical aspects require improvement.

The methodology for species-level identification of the S. thermophilus isolates is not described. It is also recommended that the article clearly highlight what was newly achieved, as it is not evident how these results are contributing to the field of study. 

Response 1: Thank you for pointing this out. Throughout the study, 16S PCR primers were employed as controls in each PCR step. This approach was chosen because 16S primers provide consistent guidance during the optimization of PCR conditions, as they are less affected (negatively) by variations in annealing temperature changes. Following the acquisition of CRISPR PCR results, sequencing was performed, and database analysis confirmed that all sequences were exclusively associated with S. thermophilus. Although previous studies have highlighted the potential of CRISPR as a tool for species-level identification, we opted to retain the 16S PCR products until the study was finalized. Consequently, the 16S PCR products were not subjected to sequencing analysis.

In light of the reviewer's feedback, we have focused on clearly highlighting the novel findings of the study and providing a more detailed explanation of how the results advance the field in the Discussion section. The revised manuscript is submitted for your evaluation.

Comments 2: The abstract outlines the main results, but it needs refinement to better explain the connection between the CRISPR profile and the various analyses of the technological properties of the starters. Is there any evidence that the incidence of CRISPR sequences directly proportional to the complementary analyses of gene presence and antimicrobial activity? This point is unclear and would be great to know. 

Response 2: Thank you for pointing this out. We agree with this comment. However, there is no data indicating a direct correlation; our aim was to investigate whether such a direct relationship could be established. We have made an effort to explore this topic further in the discussion section.

Comments 3: In the introduction, I suggest adding a sentence to mention that in-depth study of CRISPR sequences has implications for gene editing, which is also highly relevant in food technology. Lines 98 to 105 briefly touch upon this and could be moved earlier in the introduction. This suggestion stems from the fact that the first association with CRISPR is often gene editing, even though that is not the focus of this work.

Moreover, it would be good to include references to other studies that have characterized CRISPR sequences to assess immune response in bacterial strains.

Response 3: Thank you for pointing this out. We agree with this comment. Therefore, we have added a sentence to the first paragraph of the introduction page number 2,  paragraph  1, and lines 46-47. “In the context of optimizing starter cultures, modifications to adaptive immune mechanisms are extensively utilized”. 

Comments 4: Line 18: Cite "Turkey" in English instead of the original language, as well as in other instances where it appears. This will contribute to access your nation results in academia research. 

Response 4: Thank you for pointing this out. It is recommended to continue using "Türkiye", as the officially recognized name has been changed to "Türkiye" at the United Nations.

Comments 5: In the introduction section, from lines 27 to 49, there are no theoretical references. Please include citations.

Response 5: Thank you for pointing this out. We agree with this comment. Therefore, we have added citations page number 1,  paragraph  1, and lines 34. 

[1] Özcan, A., Yıbar, A., Kiraz, D. et al. Comprehensive analysis of the CRISPR-Cas systems in Streptococcus thermophilus strains isolated from traditional yogurts. Antonie van Leeuwenhoek 117, 63 (2024).

Comments 6: Lines 67-68: lactic acid bacteria are part of the bacterial group. This sentence needs to be revised, as it is unclear and requires proper referencing.

Response 6: Thank you for pointing this out. We agree with this comment. Therefore, we have deleted the sentence: “However, the immune function of this system was able to be elucidated through subsequent studies. The CRISPR systems are present in 40% of bacteria, and 90% of archaea.”. 

Comments 7: Lines 74, 80, and 81: It is suggested to select the most promising and current studies for references, as there are many included, and some may be more relevant than others.

Response 7: Thank you for pointing this out. We agree with this comment. Therefore, we have reduced the references.

  • We have deleted references:  13, 14, 15, 16, 17, 18, 19, 20, 21, 22, 23 on page number 2, paragraph  4, and line 72.
  • We have deleted references:  31, 32, 33, 34, 35, 36, 37, 38, 39, 40, 41,42 18 on page number 2, paragraph  5, and line 79.
  • We have deleted references:  15, 55, 56, 57, 58 on page number 2, paragraph  7, and line 89.

Comments 8: Line 107: Use only the abbreviation "LAB," as it was previously cited in the introduction.

Response 8: Thank you for pointing this out. We agree with your comment and have made the necessary correction on page 2, paragraph 7, line 89, where the abbreviation “LAB” is introduced for the first time. It has been consistently used throughout the text thereafter.

Comments 9: Overall, the introduction could be better structured to highlight the relevance of all the tests conducted for determining the technological properties of the strain, citing other works in the field that have been published for the same type of analysis regarding strain immunity. 

Response 9: Thank you for pointing this out. We agree with this comment. Therefore, we have added reduced current introduction and include relevant references. 

Comments 10: The sections of the introduction discussing genes are interesting but somewhat exhaustive. The research conducted is highly significant, and a lengthy section on genes is not the primary focus. Therefore, I suggest summarizing it.

Response 10: Thank you for pointing this out. We agree with this comment. Accordingly, we have summarized the information regarding the genes and incorporated some of it into the Discussion section, as suggested by the subsequent reviewer comments.

We have deleted these sentences from the introduction:

-In addition to the previously known receptor mutation and restriction modification developed against foreign genetic materials, CRISPR has been identified as the most fundamental immune structure component. 

-For CRISPR activity, CRISPR-associated (Cas) genes, which are located adjacent to CRISPR sequences and encode proteins necessary for immune response, as well as mature crRNA and tracrRNA, are required [43,44]. Acquired adaptive immunity is established by the positioning of spacer regions derived from exogenous nucleic acid (phage or plasmid) sources within the CRISPR region. The protospacer in the exogenous nucleic acid (phage or plasmid) is recognized by the Cas1 and Cas2 proteins, which exhibit recombinase and nuclease activity, and it is integrated into the closest position following the leader sequence [45]. When the bacterial cell encounters this mobile genetic material again, transcription of preCRISPR occurs from the CRISPR sequence, followed by transcription of Cas9 nucleases and tracrRNAs. 

-Currently, CRISPR technology has been used in various fields including basic biology, medical genetics, cancer research, immunology, infectious diseases, microbiology, and others. By activating, silencing, or manipulating target genes in different ways, it has enabled the acquisition of detailed data [46]. The CRISPR systems in microorganisms can be regulated in laboratory conditions to build CRISPR adaptation-mediated library manufacturing (CALM) [47]. An array of specific spacers can be assembled into a vector via Golden Gate assembly, leading to high activation of transcriptional regulation by the dCas9 system [48, 49]. 

-To illustrate, in the first step of the proteolytic system, the extracellular cell wall proteinase, prtS, is responsible for the degradation of caseins αS1 and β and is essential for processes such as yogurt fermentation. The second step is the transport of amino acids and peptides, through the cell membrane. In the third step of the proteolytic system, after amino acid and peptide transport, transported peptides are cleaved by intracellular aminopeptidases and peptidases to reveal essential amino acid residues. In total, there are 14 peptidases in the proteolytic system of S. thermophilus, five of which have different characteristics and are specific to S. thermophilus. These are PepO, PepC, PepS, PepX, and PepN [61].

-The enzyme in S. thermophilus is encoded by the gene cluster ureABC, which also includes accessory genes ureEFGD and ureI that control the biogenesis of urease. Although urease is expressed by the majority of S. thermophilus strains, some urease-negative mutants have been created that show faster acidification in milk, which makes them more appropriate for industrial yogurt production [63, 64, 65].

- In the production of dairy products, the GDH enzyme contributes to protein breakdown and the production of aromatic compounds. Lactic acid bacteria (LAB), particularly during the fermentation process, help the production of free amino acids from milk proteins and the formation of compounds that play an important role in product taste and flavor. Particularly in cheese production, the release of glutamate and its interaction with other by-products increase the characteristic flavor of the cheese [67]. Of note, S. thermophilus shows significant GDH activity, particularly compared to other LAB such as lactobacilli. These metabolic processes are essential in producing flavor compounds including alcohols, aldehydes, and carboxylic acids, contributing significantly to the sensory characteristics of dairy products. [68].

-They are acidic proteins with a low molecular mass of 7.5 kDa. [69]. These proteins can be specifically important in dairy production considering frozen starters are commonly used. 

-Study shows that S. thermophilus LMD-9 has two cold-shock protein genes (STER0879 and STER0880) which are only similar in sequence by 50%, indicating a difference in function. However, the presence of cold shock protein genes seems to have adapted the bacteria to cope with the stress during commercial production [71]. Several studies have also tested the freezing tolerance of S. thermophilus after a cold shock treatment. Overall, the results indicate that different strains and treatment conditions (temperature and time) yield variable results [72]. Nonetheless, further genomic and proteomic studies are needed to better understand their potential for cold adaptability. Cold stress is thought to be critical for the survival of frozen starter cultures and for fermentation occurring at low temperatures [73]. 

-Sigma factors do this by binding to the RNA polymerase enzyme and finding two copies as comX1 and comX2. Therefore, the foreign DNA transformation is affected positively by the comX gene presence. According to Karaman mutations in the comX gene have a strong relationship with competence frequency and conjugation capacity [74]. The comX gene is critical in terms of increasing quality and efficiency in yogurt and cheese production processes where S. thermophilus and Streptococcus salivarius bacteria are used, as it provides natural competence [75].

Comments 11: Lines 212, 303, 343, 359, 387: Correct double spacing.

Response 11: Thank you for pointing this out. We agree with this comment. Therefore, we have corrected the double spacing in the text (148, 245, 263, 271,

Comments 12: Lines 217 and 273: Species names should be italicized.

Response 12: Thank you for pointing this out. We agree with this comment. Therefore, we have corrected italics throughout the manuscript. “Highlighted with yellow on various pages.”

Comments 13: Lines 221 and 222: Use superscript for "-1" and "-8" 

Response 13: Thank you for pointing this out. Therefore, we have corrected superscripts in the text in Therefore, we have corrected the text to have superscripts for the dilutions on page 4, paragraph 2, lines 158-159.

Comments 14: The materials and methods section does not include how the strains were identified. Morphological classification alone is insufficient for species-level identification. It is necessary to describe the identification protocol for the isolated strains. Furthermore, it is expected that the items in the materials and methods follow a similar order to that described in the introduction and/or results and discussion sections. Please, check. 

Response 14:  Thank you for pointing this out. We agree with this comment. Therefore, we have corrected the order in the Introduction, Methods, Results, and Discussion sections. Throughout the study, 16S PCR primers were employed as controls in each PCR step. This approach was chosen because 16S primers provide consistent guidance during the optimization of PCR conditions, as they are less affected (negatively) by variations in annealing temperature changes. Following the acquisition of CRISPR PCR results, sequencing was performed, and database analysis confirmed that all sequences were exclusively associated with Streptococcus thermophilus. Although previous studies have highlighted the potential of CRISPR as a tool for species-level identification, we opted to retain the 16S PCR products until the study was finalized. Consequently, the 16S PCR products were not subjected to sequencing analysis. In light of the reviewer's feedback, we have focused on clearly highlighting the novel findings of the study and providing a more detailed explanation of how the results advance the field in the discussion section. The revised manuscript is submitted for your evaluation.

Comments 15: Line 228: The reference does not adhere to the journal's format.

Response 15: Thank you for pointing this out. We agree with this comment. Therefore, we have corrected the reference on page 4 paragraph 4 line 167.

Comments 16: Lines 238 and 244: Correct the formatting of the "2" in CaCl₂ and the "7" in 10⁷ cfu/mL. Please, review this throughout the text. 

Response 16: Thank you for pointing this out. We agree with this comment. Therefore, we have corrected the "2" in CaCl₂ and the "7" in 10⁷ cfu/mL on page 4, line 175. 

Comments 17: Line 242:  Bacillus cereus CCM 99 AND E. coli ATCC 25922.

Response 17: Thank you for pointing this out. We agree with this comment. Therefore, we have corrected these on page number 11,  paragraph 1, and line 308.

Comments 18: Were the proteolytic, lipolytic, antimicrobial activities, and CRISPR profile analyses conducted following any previously described protocol? Please cite accordingly.

Response 18: Thank you for pointing this out. We agree with this comment. Proteolytic, lipolytic, antimicrobial activity and CRISPR profile analyses were conducted as previously described, and the citations have been revised as requested on page 4, paragraph 6, line 189

Comments 19: Lines 249, 264, 340: Use "isolates."

Response 19: Thank you for pointing this out. We agree with this comment. Therefore, we have corrected it as isolates on page 8, in line 254. 

Comments 20: Line 252: I suggest placing all tables immediately after they are cited to facilitate reading.

Response 20: Thank you for pointing this out. We agree with this comment. Therefore, we have changed the location in the text to the appropriate place in Table 1: Line 205, Table 2: Line 215, Table 3: line 229, Table 4: Line 239, Table 5: Line 256, Table 6.: Line 258 and Table 7: Line 267 

Comments 21: Lines 274-280: Are selective media and morphological characteristics sufficient for species-level identification? What evidence supports this conclusion?

Response 21: Thank you for pointing this out. We have completed morphological characterisation but CRISPR sequence analysis confirmed that the isolates are Streptococcus thermophilus. Throughout the study, 16S PCR primers were employed as controls in each PCR step. This approach was chosen because 16S primers provide consistent guidance during the optimization of PCR conditions, as they are less affected (negatively) by variations in annealing temperature changes. Following the acquisition of CRISPR PCR results, sequencing was performed, and database analysis confirmed that all sequences were exclusively associated with Streptococcus thermophilus. Although previous studies have highlighted the potential of CRISPR as a tool for species-level identification, we opted to retain the 16S PCR products until the study was finalized. Consequently, the 16S PCR products were not subjected to sequencing analysis. In light of the reviewer's feedback, we have focused on clearly highlighting the novel findings of the study and providing a more detailed explanation of how the results advance the field in the discussion section. The revised manuscript is submitted for your evaluation.

Comments 22: Lines 283-286: The sentence is confusing; please rephrase. Also, the reference does not follow the journal's standards.

Response 22: Thank you for pointing this out. We agree with this comment. Therefore, we have deleted this detailed info from the current place and used this text in the discussion by correcting the reference format references were also corrected as ref 46 and 47 on page number 10,  paragraph 2,  and line 291. 

Comments 23: Line 296: S. thermophilus.

Response 23: Thank you for pointing this out. We agree with this comment. We have relocated this info from results to discussion based on the reviewer's requests and corrected the italics for S. thermophilus. in the context. 

Comments 24: Line 300: Were tests for antimicrobial activity conducted against S. aureus and P. aeruginosa? These are the most common for this essay, besides E. coli.

Response 24: Thank you for pointing this out. We agree with this comment. These analyses have not yet been conducted, as the reference strains are in the process of being obtained. However, the analyses are planned to be included in the thesis study.

Comments 25: Lines 331-335: The sentence is unclear; please reconstruct it.

Response 25: Thank you for pointing this out. We agree with this comment. Therefore, we have deleted this sentence. 

Comments 26: Lines 343-344: WAS positive.

Response 26: Thank you for pointing this out. We have corrected it as “was positive” on page number 9,  paragraph 2, and line 264.

Comments 27: Line 350: References in Table 2 are not in the correct journal format.

Response 27: Thank you for pointing this out. We agree with this comment. Therefore, we have corrected the table formats on page number 5,  paragraph  2, and line 215.

Comments 28: Line 352: Standardize tables font size.

Response 28: Thank you for pointing this out. We agree with this comment. Therefore, we have standardized table formats. 

Table 1: Line 205, Table 2: Line 215, Table 3: line 229, Table 4: Line 239, Table 5: Line 256, Table 6.: Line 258 and Table 7: Line 267 

Comments 29: Lines 357-362: Add a reference.

Response 29: Thank you for pointing this out. We agree with this comment. Therefore, we have added a reference. Page number 10,  paragraph 1, and line 274.

Siddiqui, S.A., Erol, Z., Rugji, J. et al. An overview of fermentation in the food industry - looking back from a new perspective. Bioresour. Bioprocess. 10, 85 (2023). https://doi.org/10.1186/s40643-023-00702-y

Comments 30: Line 364: Use the abbreviation "LAB."

Response 30: Thank you for pointing this out. We agree with this comment. Therefore, we have used the abbreviation "LAB." page number 10,  paragraph 1, and line 277.

Comments 31: Lines 378-380: The sentence raises the question of why these analyses have not yet been conducted if they are important. I suggest omitting this since it was not performed. Instead, you may state that further exploration of EPS production capabilities could be achieved through primer analysis X. 

Response 31: Thank you for pointing this out. We agree with this comment. Therefore, we have deleted the EPS analysis based on the reviewer's comments. 

-deleted form line 24/25:  and the EPS gene cluster, targeting the exopolysaccharide locus sequence

-deleted  1.6. Exopolysaccharide Locus Sequence (EPS gene): Exopolysaccharides (EPSs) are associated with the ability to modulate the mucosal immune system and, through this, possess the capacity to modulate antiviral immune responses [77]. The EPS gene cluster participates in EPS synthesis (Stingele, 1996). Secreted EPS macromolecules cause a viscous, slime-like texture that protects the cells from extreme conditions, including osmotic stress and bile [78]. The EPS, produced by LAB, is a natural alternative which can be used instead of polysaccharides such as pectin, guar gum, starch, etc. used as additives [79]. Since EPS directly affects the texture of the end product of yogurt fermentation, its production mechanism and related genes have rising prominence for research activities [77].

-deleted 2.2.2. Technologically important proteins of the Isolates: the EPS gene cluster, targeting the EPS [77] 

- EPS PCR primers deleted from Table 2

- The sentence deleted from 3.2.2. Technologically important proteins of the isolates: For the EPS gene cluster, targeting the EPS, isolates were PCR positive only one (4%) of the isolates

EPS-related information was deleted from Table 7

These sentences were deleted from the discussion: Secreted EPS macromolecules cause a viscous, slime-like texture that protects the cells from extreme conditions, including osmotic stress and bile [78]. The EPS, produced by LAB, is a natural alternative which can be used instead of polysaccharides such as pectin, guar gum, starch, etc. used as additives [79]. Since EPS directly affects the texture of the end product of yogurt fermentation, its production mechanism and related genes have rising prominence for research activities [77]. Our isolates were recorded as eps negative with selected previous studies, the expression of plasmid-encoded EPS genes was observed at higher transcription levels under optimized culture conditions [93]. Therefore, we attribute the negative EPS PCR results from our experiments conducted under non-optimized conditions to inadequate plasmid copy numbers. Additionally, apart from the EPS primers we selected, it is planned to repeat the PCR screenings using other known primers such as epsB, epsE, and epsG. In the subsequent phases of the study, characteristics such as biofilm formation [94]. EPS genes, lactose transport protein, and Streptococcal competence (com) genes, which exhibit distinct traits in the genetic domestication of S. thermophilus, and are associated with commercial starters, are planned to be analyzed using genetic methods. 

Comments 32: Line 381: Check whether "Streptococcal" should be italicized and standardize it.

Response 32: Thank you for pointing this out. We have checked and confirmed that the term "Streptococcal" should not be italicized. In scientific writing, only genus and species names are italicized.

Comments 33: Regarding the discussion and conclusion section, it appears to be lacking in depth. The main conclusion is unclear. The data need to be compared with other studies on this topic. It is not evident how your study has genuinely advanced the selection of starter cultures. You have significant results, but they need better discussion. For example, were any specific CRISPR profile present in strains with a higher number of positive technological characteristics? Are there other LAB strains with the same profiles? If so, which ones? Was this the expected outcome for your group of study? What was the most significant conclusion of the article following this characterization of the isolates? Which analyses would be most promising for selecting the most suitable isolates for application as starters?

Response 33: Thank you for pointing this out. We agree with this comment. Therefore, we have updated the discussion section according to the revisions.

Comments 34: Overall, the English language in the article could be improved by increasing formality, enhancing the use of connectors, and improving sentence transitions to elevate the quality of the material. Also, avoid the use of colloquial language Lines 362-363: In this sentence, "enriches our table" and "intestinal flora" should be written in more formal English. The same applies to lines 381-384.

Response 33: Thank you for pointing this out. We agree with this comment. Therefore, we have updated the discussion section according to the revisions. enriches our table was deleted from sentence and intestinal microbiota were used instead of intestinal flora

Reviewer 2 Report

Comments and Suggestions for Authors

GENERAL COMMENTS

The study obtained data on Clustered Regularly Interspaced Short Palindromic Repeats (CRISPR) profiles of Streptococcus thermophilus (S. thermophilus) for selecting potential starter cultures from local yogurt samples. The authors conclude that the results suggest that it is possible to choose the isolates with the desired technological characteristics, prioritizing the ones with the most diverse CRISPR profile and with native spacers for potential industrial application as a starter culture. They also consider that this study provides data for further biological studies on the impact of centuries of human domestication on evolutionary adaptations and how these microorganisms manage survival and symbiosis.

The manuscript shows an adequate experimental design. Nonetheless, it needs some major corrections.

  1. Introduction. The introduction is too long and contains a lot of general information about CRISPR. The authors should make an effort to condense the information to at least half its length.

  2. Materials and methods. The experiments are clearly described. The authors must address some minor issues as detailed in the specific comments.

  3. Results and discussion. 

    1. The manuscript emphasizes two desired characteristics in strains used as fermentation starters to produce yogurt. One is lipolytic capacity, and the other is antimicrobial activity. It is reported that none of the isolates presented these characteristics. It is only mentioned, and no discussion is made about it. However, it is concluded that the isolates can be used as fermentation starters. A thorough discussion of the results in general and of these two aspects in particular must be made.

    2. A core aspect of the work is the detection of genes. The authors mention the negative EPS PCR results, but they justify them in a very simple way, attributing them to inadequate experimentation (Line 376-378). A more in-depth discussion of this aspect should be made, or those determinations should be repeated, solving the experimental problems to detect what is sought or to demonstrate that those strains do not have that characteristic and discuss its implications.

COMMENTS ON ENGLISH

The English is good. 

SPECIFIC COMMENTS

  1. Line 217. Please write “S. thermophilus” in italics. Please ensure that all scientific names are written in italics throughout the manuscript.

  2. Line 221-222. Please use superscript where appropriate.

  3. Line 226-227. Please provide the preservation temperature.

  4. Line 244. Please use superscript where appropriate.

  5. Line 348-356. It is suggested that subsection 3.4 be removed and the tables relocated to positions close to their first mention in the text.

Author Response

November 18th, 2024

Dear Editor,

On behalf of my colleagues, I would like to thank you and the reviewers for their efforts and comments which led us to improve the scientific quality of our manuscript entitled “Determination of technological properties and CRISPR profiles of Streptococcus thermophilus isolates obtained from local yogurt samples” and for the opportunity to address the reviewers’ comments.

We thank the reviewers for their constructive comments. All relevant amendments were made accordingly as highlighted in the main manuscript in yellow color. A point-by-point response to the reviewers’ comments has been presented below. 

Kindly submitted for your consideration for publication.

Yours sincerely,

Prof. Dr. Muhammet Zeki Durak

Department of Food Engineering, 

Faculty of Chemical and Metallurgical Engineering, 

Yıldız Technical University, İstanbul, TÜRKİYE

Comments 1: Introduction. The introduction is too long and contains a lot of general information about CRISPR. The authors should make an effort to condense the information to at least half its length.

Response 1: Thank you for pointing this out. We agree with this comment. In line with your recommendations, select portions from the introduction and results sections have been integrated into the discussion to enhance the analytical depth of the manuscript. Furthermore, additional commentary has been provided to further enrich the discussion. The revised manuscript is hereby submitted for your review and evaluation.

We have deleted these sentences from the introduction:

-In addition to the previously known receptor mutation and restriction modification developed against foreign genetic materials, CRISPR has been identified as the most fundamental immune structure component. 

-For CRISPR activity, CRISPR-associated (Cas) genes, which are located adjacent to CRISPR sequences and encode proteins necessary for immune response, as well as mature crRNA and tracrRNA, are required [43,44]. Acquired adaptive immunity is established by the positioning of spacer regions derived from exogenous nucleic acid (phage or plasmid) sources within the CRISPR region. The protospacer in the exogenous nucleic acid (phage or plasmid) is recognized by the Cas1 and Cas2 proteins, which exhibit recombinase and nuclease activity, and it is integrated into the closest position following the leader sequence [45]. When the bacterial cell encounters this mobile genetic material again, transcription of preCRISPR occurs from the CRISPR sequence, followed by transcription of Cas9 nucleases and tracrRNAs. 

-Currently, CRISPR technology has been used in various fields including basic biology, medical genetics, cancer research, immunology, infectious diseases, microbiology, and others. By activating, silencing, or manipulating target genes in different ways, it has enabled the acquisition of detailed data [46]. The CRISPR systems in microorganisms can be regulated in laboratory conditions to build CRISPR adaptation-mediated library manufacturing (CALM) [47]. An array of specific spacers can be assembled into a vector via Golden Gate assembly, leading to high activation of transcriptional regulation by the dCas9 system [48, 49]. 

-To illustrate, in the first step of the proteolytic system, the extracellular cell wall proteinase, prtS, is responsible for the degradation of caseins αS1 and β and is essential for processes such as yogurt fermentation. The second step is the transport of amino acids and peptides, through the cell membrane. In the third step of the proteolytic system, after amino acid and peptide transport, transported peptides are cleaved by intracellular aminopeptidases and peptidases to reveal essential amino acid residues. In total, there are 14 peptidases in the proteolytic system of S. thermophilus, five of which have different characteristics and are specific to S. thermophilus. These are PepO, PepC, PepS, PepX, and PepN [61].

-The enzyme in S. thermophilus is encoded by the gene cluster ureABC, which also includes accessory genes ureEFGD and ureI that control the biogenesis of urease. Although urease is expressed by the majority of S. thermophilus strains, some urease-negative mutants have been created that show faster acidification in milk, which makes them more appropriate for industrial yogurt production [63, 64, 65].

- In the production of dairy products, the GDH enzyme contributes to protein breakdown and the production of aromatic compounds. Lactic acid bacteria (LAB), particularly during the fermentation process, help the production of free amino acids from milk proteins and the formation of compounds that play an important role in product taste and flavor. Particularly in cheese production, the release of glutamate and its interaction with other by-products increase the characteristic flavor of the cheese [67]. Of note, S. thermophilus shows significant GDH activity, particularly compared to other LAB such as lactobacilli. These metabolic processes are essential in producing flavor compounds including alcohols, aldehydes, and carboxylic acids, contributing significantly to the sensory characteristics of dairy products. [68].

-They are acidic proteins with a low molecular mass of 7.5 kDa. [69]. These proteins can be specifically important in dairy production considering frozen starters are commonly used. 

-Study shows that S. thermophilus LMD-9 has two cold-shock protein genes (STER0879 and STER0880) which are only similar in sequence by 50%, indicating a difference in function. However, the presence of cold shock protein genes seems to have adapted the bacteria to cope with the stress during commercial production [71]. Several studies have also tested the freezing tolerance of S. thermophilus after a cold shock treatment. Overall, the results indicate that different strains and treatment conditions (temperature and time) yield variable results [72]. Nonetheless, further genomic and proteomic studies are needed to better understand their potential for cold adaptability. Cold stress is thought to be critical for the survival of frozen starter cultures and for fermentation occurring at low temperatures [73]. 

-Sigma factors do this by binding to the RNA polymerase enzyme and finding two copies as comX1 and comX2. Therefore, the foreign DNA transformation is affected positively by the comX gene presence. According to Karaman mutations in the comX gene have a strong relationship with competence frequency and conjugation capacity [74]. The comX gene is critical in terms of increasing quality and efficiency in yogurt and cheese production processes where S. thermophilus and Streptococcus salivarius bacteria are used, as it provides natural competence [75].

Comments 2: Materials and methods. The experiments are clearly described. The authors must address some minor issues as detailed in the specific comments. The manuscript emphasizes two desired characteristics in strains used as fermentation starters to produce yogurt. One is lipolytic capacity, and the other is antimicrobial activity. It is reported that none of the isolates presented these characteristics. It is only mentioned, and no discussion is made about it. However, it is concluded that the isolates can be used as fermentation starters. A thorough discussion of the results in general and of these two aspects in particular must be made.

Response 2: Thank you for pointing this out. Lipolytic activity, proteolytic activity, and antimicrobial activity are characteristics rarely observed in Streptococcus thermophilus strains, including starter cultures, in their natural forms. In this context, it would not be entirely accurate to state that cultures lacking antimicrobial activity cannot be used as starters. Similarly, the absence of lipolytic activity in our isolates cannot definitively be said to preclude their use as starter cultures. On the other hand, the fact that all isolates exhibit proteolytic activity is potentially significant. While we do not have a clear hypothesis regarding any correlations between these traits and other characteristics screened in the study, our aim was to collect data on this topic. To our knowledge, no study has yet demonstrated a definitive correlation between acquired immunity and starter potential. Although genetic analyses have been widely conducted, we aimed to include these traits in our screening to enable comparison and further insight. In line with the requested revisions, we have made an effort to integrate the aforementioned information into the Discussion section to enrich its content and provide a more comprehensive explanation.

Comments 3: A core aspect of the work is the detection of genes. The authors mention the negative EPS PCR results, but they justify them in a very simple way, attributing them to inadequate experimentation (Line 376-378). A more in-depth discussion of this aspect should be made, or those determinations should be repeated, solving the experimental problems to detect what is sought or to demonstrate that those strains do not have that characteristic and discuss its implications.

Response 3: Thank you for pointing this out. We agree with this comment. Considering all reviewer feedback, it has been decided to remove the EPS section from the manuscript.

–deleted from lines 24/25:  and the EPS gene cluster, targeting the exopolysaccharide locus sequence

–-deleted 1.6. Exopolysaccharide Locus Sequence (EPS gene): Exopolysaccharides (EPSs) are associated with the ability to modulate the mucosal immune system and, through this, possess the capacity to modulate antiviral immune responses [77]. The EPS gene cluster participates in EPS synthesis (Stingele, 1996). Secreted EPS macromolecules cause a viscous, slime-like texture that protects the cells from extreme conditions, including osmotic stress and bile [78]. The EPS, produced by LAB, is a natural alternative which can be used instead of polysaccharides such as pectin, guar gum, starch, etc. used as additives [79]. Since EPS directly affects the texture of the end product of yogurt fermentation, its production mechanism and related genes have rising prominence for research activities [77].

–deleted 2.2.2. Technologically important proteins of the Isolates: the EPS gene cluster, targeting the EPS [77] 

–EPS PCR primers deleted from Table 2

–The sentence deleted from 3.2.2. Technologically important proteins of the isolates: For the EPS gene cluster, targeting the EPS, isolates were PCR positive only one (4%) of the isolates

–EPS-related information was deleted from Table 7

–These sentences were deleted from the discussion: Secreted EPS macromolecules cause a viscous, slime-like texture that protects the cells from extreme conditions, including osmotic stress and bile [78]. The EPS, produced by LAB, is a natural alternative which can be used instead of polysaccharides such as pectin, guar gum, starch, etc. used as additives [79]. Since EPS directly affects the texture of the end product of yogurt fermentation, its production mechanism and related genes have rising prominence for research activities [77]. Our isolates were recorded as eps negative with selected previous studies, the expression of plasmid-encoded EPS genes was observed at higher transcription levels under optimized culture conditions [93]. Therefore, we attribute the negative EPS PCR results from our experiments conducted under non-optimized conditions to inadequate plasmid copy numbers. Additionally, apart from the EPS primers we selected, it is planned to repeat the PCR screenings using other known primers such as epsB, epsE, and epsG. In the subsequent phases of the study, characteristics such as biofilm formation [94]. EPS genes, lactose transport protein, and Streptococcal competence (com) genes, which exhibit distinct traits in the genetic domestication of S. thermophilus, and are associated with commercial starters, are planned to be analyzed using genetic methods. 

Comments 4: Line 217. Please write “S. thermophilus” in italics. Please ensure that all scientific names are written in italics throughout the manuscript.

Response 4: Thank you for pointing this out. We agree with this comment. Therefore, we have corrected italics throughout the manuscript. “Highlighted with yellow on various pages.”

Comments 5: Line 221-222. Please use superscript where appropriate.
Response 5: Thank you for pointing this out. We agree with this comment. Therefore, we have corrected the text to have superscript for the dilutions on page 4, paragraph 2, lines 158-159.

Comments 6: Line 226-227. Please provide the preservation temperature.
Response 6: Thank you for pointing this out. We agree with this comment. Therefore, we have provided a preservation temperature in page number 4,  paragraph 2, and line 164.  

Comments 7: Line 244. Please use superscript where appropriate.
Response 7: Thank you for pointing this out. We agree with this comment. Therefore, we have corrected superscripts in the text. Therefore, we have corrected the text to have superscripts for the dilutions on page 4, paragraph 2, lines 158-159.

Comments 8: Line 348-356. It is suggested that subsection 3.4 be removed and the tables relocated to positions close to their first mention in the text.

Response 8: Thank you for pointing this out. We agree with this comment. Therefore, we have removed it and relocated the tables to positions close to their first mention in the text. Table 1: Line 205, Table 2: Line 215, Table 3: line 229, Table 4: Line 239, Table 5: Line 256, Table 6.: Line 258 and Table 7: Line 267

Reviewer 3 Report

Comments and Suggestions for Authors

microorganisms-3318490-peer-review-v1

Ln22: genes needs to be in italics.

Ln107: Please, introduced abbreviation LAB earlier in the paper, when for the first time term was provided.

Introduction was presented well and state principal points on following/explored by the paper research.

In some cases, authors have provided an extended list of references. Maybe it is possible a reduction of numbers and focus on principle review papers reporting on the subject will be appropriate.

Introduction can be reduced, and specially information about different genes presented can be move to the discussion section. In my opinion will be more appropriate.

Ln221,222: Please, more to exponential position in dilutions levels information.

Ln234-240: Please, provide reference and a bit more details

Ln241: Please, provide reference. In this performed experiment, you have only observed cell to cell inhibitory activity.

Ln264-269: Bit more details needs to be provided.

Ln281-305: The text was provided, but not reference to the respected tables where results are summarized. Please, maybe help form more experienced colleges can be a good option to improve presentation of the manuscript.

Please be sure that in results section you can focus on results and not discussion. Example, on 3.1.4. Please, provide exact results what you have observed. 3.1.3. What was results? What strains were and what were not showing proteolytic and lipolytic activity. Maybe table with summary of the results can be good option.

Ln363: Maybe "intestinal microbiota" will be a more appropriate term to be used.

Discussion is very basic. Authors need to improve this section of the manuscript. As was previously mentioned some parts can be move from the introduction to the discussion section.

References need to be according to the instructions from the journal.

Author Response

November 18th, 2024

Dear Editor,

On behalf of my colleagues, I would like to thank you and the reviewers for their efforts and comments which led us to improve the scientific quality of our manuscript entitled “Determination of technological properties and CRISPR profiles of Streptococcus thermophilus isolates obtained from local yogurt samples” and for the opportunity to address the reviewers’ comments.

We thank the reviewers for their constructive comments. All relevant amendments were made accordingly as highlighted in the main manuscript in yellow color. A point-by-point response to the reviewers’ comments has been presented below.

Kindly submitted for your consideration for publication.

Yours sincerely,

Prof. Dr. Muhammet Zeki Durak

Department of Food Engineering,

Faculty of Chemical and Metallurgical Engineering,

Yıldız Technical University, İstanbul, TÜRKİYE

Comments 1: Ln22: genes need to be in italics.

Response 1: Thank you for pointing this out. We agree with this comment. Therefore, we have corrected genes that need to be in italics in the article. “Highlighted with yellow in various pages.”

Comments 2: Ln107: Please, introduce abbreviation LAB earlier in the paper, when for the first time term was provided.

Response 2: Thank you for pointing this out. We agree with this comment. Therefore, we have updated it as suggested. We have used the abbreviation lactic acid bacteria (LAB) in the first correct line in the sentence “To date, the CRISPR profiles and genes associated with technological traits in various lactic acid bacteria (LAB) have been thoroughly investigated [50, 51, 52, 53, 54] page number 1,  paragraph 7, and line 88-89.

Comments 3: Introduction was presented well and state principal points on following/explored by the paper research. In some cases, authors have provided an extended list of references. Maybe it is possible a reduction of numbers and focus on principle review papers reporting on the subject will be appropriate.

Response 3: Thank you for pointing this out. We agree with this comment. Therefore, we have reduced the references focusing on the core articles.  

We have deleted references 3, 14, 15, 16, 17, 18, 19, 20, 21, 22, 23 on page number 2, paragraph  4, and line 72.
We have deleted references  31, 32, 33, 34, 35, 36, 37, 38, 39, 40, 41,42 18 on page number 2, paragraph  5, and line 79.
We have deleted references 15, 55, 56, 57, 58 page number 2, paragraph  7, and line 89.

Comments 4: Introduction can be reduced, and specially information about different genes presented can be move to the discussion section. In my opinion it will be more appropriate.

Response 4: Thank you for pointing this out. We agree with this comment. Therefore, we have reduced the introduction to information regarding different genes and moved to the discussion section as suggested. 

We have deleted  these sentences from the introduction:

-In addition to the previously known receptor mutation and restriction modification developed against foreign genetic materials, CRISPR has been identified as the most fundamental immune structure component. 

-For CRISPR activity, CRISPR-associated (Cas) genes, which are located adjacent to CRISPR sequences and encode proteins necessary for immune response, as well as mature crRNA and tracrRNA, are required [43,44]. Acquired adaptive immunity is established by the positioning of spacer regions derived from exogenous nucleic acid (phage or plasmid) sources within the CRISPR region. The protospacer in the exogenous nucleic acid (phage or plasmid) is recognized by the Cas1 and Cas2 proteins, which exhibit recombinase and nuclease activity, and it is integrated into the closest position following the leader sequence [45]. When the bacterial cell encounters this mobile genetic material again, transcription of preCRISPR occurs from the CRISPR sequence, followed by transcription of Cas9 nucleases and tracrRNAs. 

-Currently, CRISPR technology has been used in various fields including basic biology, medical genetics, cancer research, immunology, infectious diseases, microbiology, and others. By activating, silencing, or manipulating target genes in different ways, it has enabled the acquisition of detailed data [46]. The CRISPR systems in microorganisms can be regulated in laboratory conditions to build CRISPR adaptation-mediated library manufacturing (CALM) [47]. An array of specific spacers can be assembled into a vector via Golden Gate assembly, leading to high activation of transcriptional regulation by the dCas9 system [48, 49]. 

-To illustrate, in the first step of the proteolytic system, the extracellular cell wall proteinase, prtS, is responsible for the degradation of caseins αS1 and β and is essential for processes such as yogurt fermentation. The second step is the transport of amino acids and peptides, through the cell membrane. In the third step of the proteolytic system, after amino acid and peptide transport, transported peptides are cleaved by intracellular aminopeptidases and peptidases to reveal essential amino acid residues. In total, there are 14 peptidases in the proteolytic system of S. thermophilus, five of which have different characteristics and are specific to S. thermophilus. These are PepO, PepC, PepS, PepX, and PepN [61].

-The enzyme in S. thermophilus is encoded by the gene cluster ureABC, which also includes accessory genes ureEFGD and ureI that control the biogenesis of urease. Although urease is expressed by the majority of S. thermophilus strains, some urease-negative mutants have been created that show faster acidification in milk, which makes them more appropriate for industrial yogurt production [63, 64, 65].

- In the production of dairy products, the GDH enzyme contributes to protein breakdown and the production of aromatic compounds. Lactic acid bacteria (LAB), particularly during the fermentation process, help the production of free amino acids from milk proteins and the formation of compounds that play an important role in product taste and flavor. Particularly in cheese production, the release of glutamate and its interaction with other by-products increase the characteristic flavor of the cheese [67]. Of note, S. thermophilus shows significant GDH activity, particularly compared to other LAB such as lactobacilli. These metabolic processes are essential in producing flavor compounds including alcohols, aldehydes, and carboxylic acids, contributing significantly to the sensory characteristics of dairy products. [68].

-They are acidic proteins with a low molecular mass of 7.5 kDa. [69]. These proteins can be specifically important in dairy production considering frozen starters are commonly used. 

-Study shows that S. thermophilus LMD-9 has two cold-shock protein genes (STER0879 and STER0880) which are only similar in sequence by 50%, indicating a difference in function. However, the presence of cold shock protein genes seems to have adapted the bacteria to cope with the stress during commercial production [71]. Several studies have also tested the freezing tolerance of S. thermophilus after a cold shock treatment. Overall, the results indicate that different strains and treatment conditions (temperature and time) yield variable results [72]. Nonetheless, further genomic and proteomic studies are needed to better understand their potential for cold adaptability. Cold stress is thought to be critical for the survival of frozen starter cultures and for fermentation occurring at low temperatures [73]. 

-Sigma factors do this by binding to the RNA polymerase enzyme and finding two copies as comX1 and comX2. Therefore, the foreign DNA transformation is affected positively by the comX gene presence. According to Karaman mutations in the comX gene have a strong relationship with competence frequency and conjugation capacity [74]. The comX gene is critical in terms of increasing quality and efficiency in yogurt and cheese production processes where S. thermophilus and Streptococcus salivarius bacteria are used, as it provides natural competence [75].

Comments 5: Ln221,222: Please, more to exponential position in dilutions levels information.

Response 5: Thank you for pointing this out. We agree with this comment. Therefore, we have corrected the text to have superscript for the dilutions on page 4, paragraph 2, lines 158-159.

Comments 6: Ln234-240: Please, provide reference and a bit more details

Response 6: Thank you for pointing this out. We agree with this comment. Therefore, we have provided the required info and reference according to your correction. 

–Proteolytic: (Almeida Júnior et al., 2015) in page number 3,  paragraph  3, and line  173.

Almeida Júnior, W. L. G., da Silva Ferrari, I., de Souza, J. V., da Silva, C. D. A., da Costa, M. M., & Dias, F. S. (2015). Characterization and evaluation of lactic acid bacteria isolated from goat milk. Food Control, 53, 96–103. https://doi.org/10.1016/ j.foodcont.2015.01.013 

–Lipolytic activity: (Ateşlier & Metin, 2006)  page number 3,  paragraph  3, and line  177.

Ateşlier, Z. B. B., & Metin, K. (2006). Production and partial characterization of a novel thermostable esterase from a thermophilic Bacillus sp. Enzyme and Microbial Technology, 38, 628–635. https://doi.org/10.1016/j.enzmictec.2005.07.015

Comments 7: Ln241: Please, provide reference. In this performed experiment, you have only observed cell to cell inhibitory activity.

Response 7: Thank you for pointing this out. We agree with this comment. Therefore, we have provided the required info and reference according to your correction on page number 3,  paragraph  4, and line  180. 

–Akpınar, A., Yerlikaya, O., & Kılıç, S. (2011). Antimicrobial activity and antibiotic resistance of Lactobacillus delbrueckii ssp. bulgaricus and Streptococcus thermophilus strains isolated from Turkish homemade yoghurts. https://doi.org/10.5897/AJMR10.835 

Comments 8: Ln264-269: Bit more details needs to be provided.

Response 8: Thank you for pointing this out. We agree with this comment. Therefore, we have provided detailed info as requested. 

We have added the sentences below:  

Page number 6,  paragraph  3, and lines 270-272. Acquired adaptive immunity is established by the positioning of spacer regions derived from exogenous nucleic acid (phage or plasmid) sources within the CRISPR region. 

Page number 6,  paragraph  3, and lines 275-277  The sequences were saved as native sequences, except when they matched the sequences previously recorded in the database.  

Comments 9: Ln281-305: The text was provided, but not reference to the respected tables where results are summarized. Please, maybe help form more experienced colleges can be a good option to improve presentation of the manuscript.

Response 9: Thank you for pointing this out. Regarding the use of a table in 3.1.2 we have updated as requested in page 7 paragraph 1 line 233. For proteolytic and lypolitic activity results we agree with your comment,  it could enhance the clarity and accessibility of the results for the reader. However, to optimize space, we have summarized the findings with a general statement indicating that all isolates exhibited proteolytic activity (3.1.3), whereas none displayed lipolytic activity (3.1.3) and antimicrobial activity (3.1.4). If deemed necessary for a more comprehensive presentation, a table can be provided to supplement this information. 

Comments 10: Please be sure that in results section you can focus on results and not discussion. Example, on 3.1.4. Please, provide exact results what you have observed. 3.1.3. What was results? What strains were and what were not showing proteolytic and lipolytic activity. Maybe table with summary of the results can be good option.

Response 10: Thank you for pointing this out. We agree with this comment. In accordance with your request, certain interpretations previously included in the results section have been relocated to the discussion section for a more in-depth analysis.

Regarding the use of a table, it could enhance the clarity and accessibility of the results for the reader. However, to optimize space, we have summarized the findings with a general statement indicating that all isolates exhibited proteolytic activity (3.1.3), whereas none displayed lipolytic activity (3.1.3) and antimicrobial activity (3.1.4). If deemed necessary for a more comprehensive presentation, a table can be provided to supplement this information.

We have deleted these sentences from the related sections : 

–3.1.2. Acid production ability section:

The acidifying capacity of LAB is important for the production of aroma and texture formation in dairy products. Industrial yogurt production is based on the symbiosis between S. thermophilus and Lb. bulgaricus, the equilibrium ratio is adjusted to 1:1 or 2:3 to distribute the colloidal calcium phosphate to complete dissolution, caseins are dissolved, and the yogurt gel matrix is obtained within 282 to 330 min (de Brabandereand de Baerdemaeker, 1999, Uzunsoy et al., 2023). 

–3.1.3. Proteolytic and lipolytic activity 

Both proteolytic and lipolytic enzyme activity in extracellular formation is a valuable feature for the texture and aroma production of all LAB in dairy products. S. thermophilus is known for its weekly proteolytic and lipolytic activity. 

–3.1.4. Antimicrobial activity

The antimicrobial activity of LAB is essential for long-term preservation and increasing probiotic properties of fermented products particularly originated from bacteriocins, organic acids, and hydrogen peroxide.  It is known that antimicrobial activity is less common in S. thermophilus compared to other LAB [82]. 

–3.2.1. CRISPR Profiles of the Isolates

The relationship between ecological factors and the evolution of microbial immune strategies and the increasing diversity of known prokaryotic defense systems has been investigated [83, 84]. Currently, it has been well established that closely examining the evolutionary relationships of prokaryotes is possible by tracing the cas genes and spacer sequences hidden within CRISPR profiles. Clustering based on the spacers of CRISPR arrays were performed previously [26, 51]. It has been documented that multiple types of CRISPR can coexist in S. thermophilus (https://crispr.i2bc.paris-saclay.fr/). Additionally, CRISPR I and CRISPR III, which belong to Csn type II-A, have more conserved cas genes, and spacer regions, compared to other CRISPR types [85, 49]. Notably, the success of CRISPR I in integrating new spacer sequences is evident. It has been also shown that CRISPR I exhibits the widest distribution [48, 86, 49]. The CRISPR II/Cas is classified within type III-A and is known for having less conserved DNA sequences. The presence of cas genes and repeat-spacer regions on both sides of the cassette is a distinguishing feature, while the relatively lower percentage of repeat-spacer regions and a higher proportion of cas genes compared to other CRISPR types makes the integration of new spacers more difficult. The formation of CRISPR II suggests that it is likely a degenerated fragment originating from a Gram-positive ancestor [26, 49]. It has been demonstrated that type III-A immunity can provide conditional tolerance to "non-self" genetic elements in this case, temperate phages prevent lytic infection, but tolerate lysogenization by temperate phages. Temperate phages can integrate into the bacterial chromosome while also carrying genes that provide an advantage to the host. The concept of conditional tolerance to non-self allows the gene pool to diversify and differentiate [19 ,21, 22, 23]. CRISPR IV is part of the type I-E system and is rarely observed as a CRISPR I profile. The presence of multiple cas modules upstream of the spacer region alters the transcription process CRISPR IV. Antiviral defense is provided due to the significantly unregulated expression of the gene encoding the cascade complex [87, 26, 49, 88, 89]. 

Comments 11: Ln363: Maybe "intestinal microbiota" will be a more appropriate term to be used.

Response 11: Thank you for pointing this out. We agree with this comment. Therefore, we have corrected it to “intestinal microbiota” on page number 10,   paragraph 1, and line 276.

Comments 12: Discussion is very basic. Authors need to improve this section of the manuscript. As was previously mentioned some parts can be move from the introduction to the discussion section.

Response 12: Thank you for pointing this out. We agree with this comment. In line with your recommendations, select portions from the introduction and results sections have been integrated into the discussion to enhance the analytical depth of the manuscript. Furthermore, additional commentary has been provided to further enrich the discussion. The revised manuscript is hereby submitted for your review and evaluation.

We have deleted these sentences from the introduction:

–In addition to the previously known receptor mutation and restriction modification developed against foreign genetic materials, CRISPR has been identified as the most fundamental immune structure component. 

-For CRISPR activity, CRISPR-associated (Cas) genes, which are located adjacent to CRISPR sequences and encode proteins necessary for immune response, as well as mature crRNA and tracrRNA, are required [43,44]. Acquired adaptive immunity is established by the positioning of spacer regions derived from exogenous nucleic acid (phage or plasmid) sources within the CRISPR region. The protospacer in the exogenous nucleic acid (phage or plasmid) is recognized by the Cas1 and Cas2 proteins, which exhibit recombinase and nuclease activity, and it is integrated into the closest position following the leader sequence [45]. When the bacterial cell encounters this mobile genetic material again, transcription of preCRISPR occurs from the CRISPR sequence, followed by transcription of Cas9 nucleases and tracrRNAs. 

-Currently, CRISPR technology has been used in various fields including basic biology, medical genetics, cancer research, immunology, infectious diseases, microbiology, and others. By activating, silencing, or manipulating target genes in different ways, it has enabled the acquisition of detailed data [46]. The CRISPR systems in microorganisms can be regulated in laboratory conditions to build CRISPR adaptation-mediated library manufacturing (CALM) [47]. An array of specific spacers can be assembled into a vector via Golden Gate assembly, leading to high activation of transcriptional regulation by the dCas9 system [48, 49]. 

-To illustrate, in the first step of the proteolytic system, the extracellular cell wall proteinase, prtS, is responsible for the degradation of caseins αS1 and β and is essential for processes such as yogurt fermentation. The second step is the transport of amino acids and peptides, through the cell membrane. In the third step of the proteolytic system, after amino acid and peptide transport, transported peptides are cleaved by intracellular aminopeptidases and peptidases to reveal essential amino acid residues. In total, there are 14 peptidases in the proteolytic system of S. thermophilus, five of which have different characteristics and are specific to S. thermophilus. These are PepO, PepC, PepS, PepX, and PepN [61].

-The enzyme in S. thermophilus is encoded by the gene cluster ureABC, which also includes accessory genes ureEFGD and ureI that control the biogenesis of urease. Although urease is expressed by the majority of S. thermophilus strains, some urease-negative mutants have been created that show faster acidification in milk, which makes them more appropriate for industrial yogurt production [63, 64, 65].

- In the production of dairy products, the GDH enzyme contributes to protein breakdown and the production of aromatic compounds. Lactic acid bacteria (LAB), particularly during the fermentation process, help the production of free amino acids from milk proteins and the formation of compounds that play an important role in product taste and flavor. Particularly in cheese production, the release of glutamate and its interaction with other by-products increase the characteristic flavor of the cheese [67]. Of note, S. thermophilus shows significant GDH activity, particularly compared to other LAB such as lactobacilli. These metabolic processes are essential in producing flavor compounds including alcohols, aldehydes, and carboxylic acids, contributing significantly to the sensory characteristics of dairy products. [68].

-They are acidic proteins with a low molecular mass of 7.5 kDa. [69]. These proteins can be specifically important in dairy production considering frozen starters are commonly used. 

-Study shows that S. thermophilus LMD-9 has two cold-shock protein genes (STER0879 and STER0880) which are only similar in sequence by 50%, indicating a difference in function. However, the presence of cold shock protein genes seems to have adapted the bacteria to cope with the stress during commercial production [71]. Several studies have also tested the freezing tolerance of S. thermophilus after a cold shock treatment. Overall, the results indicate that different strains and treatment conditions (temperature and time) yield variable results [72]. Nonetheless, further genomic and proteomic studies are needed to better understand their potential for cold adaptability. Cold stress is thought to be critical for the survival of frozen starter cultures and for fermentation occurring at low temperatures [73]. 

-Sigma factors do this by binding to the RNA polymerase enzyme and finding two copies as comX1 and comX2. Therefore, the foreign DNA transformation is affected positively by the comX gene presence. According to Karaman mutations in the comX gene have a strong relationship with competence frequency and conjugation capacity [74]. The comX gene is critical in terms of increasing quality and efficiency in yogurt and cheese production processes where S. thermophilus and Streptococcus salivarius bacteria are used, as it provides natural competence [75].

Comments 13: References need to be according to the instructions from the journal.

Response 13: Thank you for pointing this out. We agree with this comment. Therefore, we have corrected the reference section and reduced citations.

We have deleted references:  13, 14, 15, 16, 17, 18, 19, 20, 21, 22, 23 on page number 2, paragraph  4, and line 72.]
We have deleted references:  31, 32, 33, 34, 35, 36, 37, 38, 39, 40, 41,42 18 on page number 2, paragraph  5, and line 79.
We have deleted references:  15, 55, 56, 57, 58 page number 2, paragraph  7, and line 89.

Round 2

Reviewer 2 Report

Comments and Suggestions for Authors

Thank you for addressing the comments and suggestions.

Reviewer 3 Report

Comments and Suggestions for Authors

Some references are still not in right format, missing italics, etc. But this can be set at later stage if the paper will be approved by the editor.